

# A numerical approach for calculating exact non-adiabatic terms in quantum dynamics

Ewen D. C. Lawrence⋆, Sebastian F. J. Schmid, Ieva Čepaitė,
Peter Kirton and Callum W. Duncan

Department of Physics, SUPA and University of Strathclyde, Glasgow, United Kingdom

⋆ ewen.lawrence@strath.ac.uk

## Abstract

Understanding how non-adiabatic terms affect quantum dynamics is fundamental to improving various protocols for quantum technologies. We present a novel approach to computing the Adiabatic Gauge Potential (AGP), which gives information on the non-adiabatic terms that arise from time dependence in the Hamiltonian. Our approach uses commutators of the Hamiltonian to build up an appropriate basis of the AGP, which can be easily truncated to give an approximate form when the exact result is intractable. We use this approach to study the AGP obtained for the transverse field Ising model on a variety of graphs, showing how the different underlying graph structures can give rise to very different scaling for the number of terms required in the AGP.

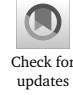
# 1 Introduction

The adiabatic approximation [1,2] states that, for a sufficiently slowly varying Hamiltonian, an initial eigenstate will remain in a corresponding eigenstate of the time-dependent problem. This approximation forms the backbone for many current methods in quantum technologies, including in adiabatic quantum computing [3–5], annealing [6,7], simulation [8–10], and the application of quantum gates [11]. The validity of the adiabatic approximation hinges on time-variations of the Hamiltonian being slow [2,5,12]. The relevant time scale for this is set by the inverse size of the gaps in its spectrum. In quantum many-body systems, gaps across transition regions are known to scale inversely with the number of degrees of freedom, forcing arbitrarily slow time-dependence in order to stay in the adiabatic regime. This has resulted in the development of a plethora of techniques to control quantum systems and achieve desired outcomes without the adiabatic approximation, leading to the development of shortcuts to adiabaticity [13,14], quantum optimal control [15–18], and diabatic quantum annealing [19]. Note that the adiabatic approximation can also be defined without the requirement that any spectral gaps exist [20,21].

For quantum many-body systems, knowing for what time-scales the adiabatic approximation breaks down is not a simple task, due to the complexity of solving the time-independent Schrödinger equation. If the Hamiltonian changes too fast, diabatic excitations across the gaps in the spectrum are possible and the definition of adiabaticity (following of a corresponding eigenstate) is violated. The approach of counterdiabatic driving [22–24] introduces additional driving terms to counter these diabatic excitations so that the adiabatic condition is enforced as the solution of the time-dependent Schrödinger equation in arbitrarily fast time. However, doing this exactly requires knowledge of the eigenstates, again requiring the solution of the time-independent Schrödinger equation. As this is beyond the reach of current computers in many scenarios, especially for more than just the ground state, a new approach to adiabaticity and counterdiabatic driving needs to be developed.

Recently, an approach was introduced which defines the diabatic excitations through the adiabatic gauge potential (AGP) which can be found variationally using the principle of least action [25,26]. The exact AGP can be found without this variational approach, but this again leads back to effectively solving the Schrödinger equation. The variational approach allows for approximate counterdiabatic drives to be constructed which can take into account requirements of practical implementation, e.g., that the control terms are local. As a result, the concept of the AGP has been applied to construct approximate counterdiabatic driving protocols for a variety of quantum many-body models, including to inform numerical optimal control [27–30], as inspiration for machine learning methods [31], to improve quantum annealing protocols [32–35], to improve state preparation [36,37], and to realise experimental demonstrations [38,39].

The AGP provides a lot of information about the dynamics of the quantum system [26], and its ability to probe physical properties of interest is still being studied. Recently, it has been shown that the norm of the AGP could provide an accurate measure of quantum phase transitions for simple models [40] and there have also been studies into its use as a measure of quantum chaos [41–43]. The AGP can be used to find optimal angles for the Quantum Approximate Optimisation Algorithm (QAOA) in a way that incorporates the suppression of diabatic losses into the Trotter error induced by a finite number of variational steps [44–46]. It has also been shown that the AGP can be utilised for calculating variational Schrieffer-Wolff transformations for the calculation of many-body dynamics [47].

In this work, we present a new, efficient numerical approach for computing the AGP which combines ideas from Refs. [25] and [48], along with the algebraic approach of Ref. [40]. Our new method can generate an approximation of the AGP to arbitrary order, while allowing for controlled truncation as necessary. It does this by exploiting symmetries, along with the relevant mathematical structure of the approximate methods. This allows us to generate a full operator basis for the AGP and to variationally determine the time-dependent coefficients for each operator in this basis. We use this new method to study the transverse field Ising model on arbitrary graphs [49] and explore how the graph connectivity affects the structure of the corresponding AGP operators and their norms. This class of Hamiltonians has been used extensively to encode solutions to combinatorial problems [44, 50, 51] and can provide insights into the behaviour of diabatic effects in many-body systems due to its simplicity and flexible structure. Within this class of problems, we explore the differences between the all-to-all and LMG models. Whilst these models share the same ground state manifold, extra steps are required to convert between the two models for the full hilbert space.

## 2 The adiabatic gauge potential

We will consider systems described by a Hamiltonian $H(\lambda(t))$ whose time-dependence is encoded in the parameter $\lambda(t)$. Here we will limit the discussion to cases where a single parameter varies in time, but the approaches discussed are straightforward to generalise to multiple parameters. The dynamics of the state, $|\Psi(t)\rangle$, of the system are governed by the time-dependent Schrödinger equation

$$i\hbar\frac{d}{dt}|\Psi(t)\rangle = H(\lambda(t))|\Psi(t)\rangle . \tag{1}$$

The instantaneous eigenstates of the Hamiltonian are defined via the time-independent Schrödinger equation

$$H(\lambda(t))|n(\lambda(t))\rangle = E_n(\lambda(t))|n(\lambda(t))\rangle , \tag{2}$$

where $|n(\lambda(t))\rangle$ and $E_n(\lambda(t))$, are the eigenstates and eigenenergies respectively. Moving forward, we will drop the explicit time dependence from the parameter in our notation for ease of reading.

In many cases, we wish to simply follow a particular eigenstate, commonly the ground state, which can be accomplished by the use of the adiabatic theorem. This states that if we initialise the system in an eigenstate and the parameters of the Hamiltonian are changed slowly enough, then the system will remain in the corresponding eigenstate of the modulated Hamiltonian up to a phase [1,2]. As discussed in Sec. 1, the slow time scale for the adiabatic condition is set by the size of the gaps in the spectrum. For example, a harmonic oscillator with frequency $\omega$ will have gaps of size $\hbar\omega$ and the adiabatic condition will be given by $|\dot{\omega}| \ll \omega^2$.

In general, the standard adiabatic condition can be stated as [52]

$$i\hbar\dot{\lambda}\frac{\langle m(\lambda)|\,\partial_\lambda H(\lambda)\,|n(\lambda)\rangle}{(E_m(\lambda)-E_n(\lambda))^2} \ll 1\,, \tag{3}$$

with $|m(\lambda)\rangle$ the eigenstate closest in energy to the eigenstate $|n(\lambda)\rangle$. However, this puts a limit on the speed of the process, in general, the parameters must be varied over an infinite time to allow the state to perfectly follow the required eigenstate.

To understand the adiabticity of a system we can transform into the co-moving frame. In general when transforming equations of motion between non-stationary frames, an extra term known as a gauge potential appears. This ensures the laws of physics are the same in all frames. A simple example of this effect is a person sitting on a merry-go-round, where they experience a fictitious centrifugal force pushing them outwards. However, an external observer would not be able to detect this force, as it arises from the gauge potential in the rotating frame, ensuring they both observe the same dynamics. The AGP is the gauge potential which results from the special transformation between the Hamiltonian in the original reference frame and the frame which co-moves with the parameter change, i.e., the frame in which the Hamiltonian is diagonal at all times. We know that the physics described by these two frames needs to be the same, and the AGP is what enforces this, as the Hamiltonian in the co-moving frame, $\tilde{H}(\lambda)$, only affects the phase between different eigenstates. We can write this transformation as

$$H(\lambda) \to \tilde{H}(\lambda) - \dot{\lambda}\tilde{\mathcal{A}}_\lambda\,, \tag{4}$$

with all diabatic excitations being generated by $\dot{\lambda}\tilde{\mathcal{A}}_\lambda$, and $\tilde{\mathcal{A}}_\lambda$ denoting the AGP in the co-moving frame which is time-dependent only through the parameter $\lambda$. Since the diabatic terms are proportional to $\dot{\lambda}$ we can immediately reconcile this transformation with the long time adiabatic limit by taking $\dot{\lambda} \to 0$, where the system evolves only under $\tilde{H}(\lambda)$, which is diagonal.

The AGP encodes information on both the spectrum and the allowed diabatic transitions. Therefore, knowledge of the AGP can be used to counter the diabatic terms [25], by adding it to the original Hamiltonian

$$H_{cd}(\lambda) = H(\lambda) + \dot{\lambda}\mathcal{A}_\lambda\,, \tag{5}$$

then we see that by transforming to the co-moving frame

$$H_{cd}(\lambda) \to \tilde{H}(\lambda) + \dot{\lambda}\tilde{\mathcal{A}}_\lambda - \dot{\lambda}\tilde{\mathcal{A}}_\lambda = \tilde{H}(\lambda)\,, \tag{6}$$

the diabatic terms are cancelled and we recover the adiabatic limit *without* requiring $\dot{\lambda} \to 0$. As the dynamics are the same in all frames, this means all diabatic terms are countered, leading to adiabatic dynamics for any $\dot{\lambda}$, i.e., we have constructed a counterdiabatic, or transitionless, drive.

We reiterate that the AGP encodes information about the diabatic transitions caused by modulation of $\lambda$. This is useful even in scenarios where we do not wish to enforce adiabatic dynamics in arbitrarily fast time. For example, we can define the magnitude of the AGP as

$$||\mathcal{A}_\lambda||^2 = \frac{\text{Tr}[\mathcal{A}_\lambda^2]}{\dim H_\lambda}\,, \tag{7}$$

giving a measure of how 'adiabatic' a dynamical path is. The definition normalises the result by the Hilbert space dimension to allow comparisons between different system sizes.

## 2.1 Exact AGP

We have shown that the AGP can be useful for countering and quantifying diabatic terms. This raises the question: How do we compute the AGP? The operator form of the AGP in the static reference frame can be written as [26]

$$\mathcal{A}_\lambda = i\partial_\lambda \,. \tag{8}$$

Note from here on we will work in natural units where $\hbar = 1$. Whilst this is an exact form of the AGP, as with any operator, to perform computations we need to represent it in a particular basis. The simplest approach is to use the eigenbasis of the Hamiltonian, which has the orthogonality condition

$$\langle m(\lambda)| H(\lambda) |n(\lambda)\rangle = 0 \,, \quad \text{for} \quad n \neq m \,. \tag{9}$$

By taking the partial differential of this condition with respect to $\lambda$, we can rearrange to get the matrix elements of the AGP

$$\langle m(\lambda)| \mathcal{A}_\lambda |n(\lambda)\rangle = i \frac{\langle m(\lambda)| \partial_\lambda H(\lambda) |n(\lambda)\rangle}{E_n(\lambda) - E_m(\lambda)} \,, \tag{10}$$

which is similar to the adiabatic condition given by Eq. (3). While this form can in principle be computed, it requires diagonalization of the Hamiltonian for every value of $\lambda$. For simple Hamiltonians where analytical solutions of the eigenstates and eigenenergies can be found, this is possible, e.g., harmonic oscillators [53] or the transverse field Ising model [54, 55].

We want to be able to derive the AGP in an arbitrary stationary basis, avoiding costly diagonalization. It can be shown that there is a basis independent condition which is satisfied by the exact AGP [26]

$$[H(\lambda), G(\mathcal{A}_\lambda)] = 0 \,, \tag{11}$$

where

$$G(\mathcal{A}_\lambda) = \partial_\lambda H(\lambda) - i[H(\lambda), \mathcal{A}_\lambda] \,. \tag{12}$$

As such the AGP can be expressed in any basis, and then its accuracy checked by computing if the above quantity is zero. This condition can be converted into a matrix equation and used to solve for the AGP algebraically [40], although this quickly becomes intractable for large or complex systems.

## 2.2 Approximate AGP

Sometimes it is not possible to compute the full AGP due to the size of the Hilbert space or complexity of the Hamiltonian. It is thus useful to define a set of operators to give an Ansatz for an approximate AGP. We can write this Ansatz as

$$\chi_\lambda = \sum_k \alpha_k(\lambda)\mathcal{O}_k \,, \tag{13}$$

with coefficients $\alpha_k(\lambda)$ and $\mathcal{O}_k$ denoting the $k$th operator in an arbitrary set. The choice for the best operator basis is problem specific, and can be very difficult to figure out *a priori*. In Ref. [25] a variational approach was developed to optimise the coefficients for each operator to obtain their approximate contribution to the exact AGP. This involves defining an appropriate action

$$S(\chi_\lambda) = \text{Tr}\left[G(\chi_\lambda)^2\right] \,, \tag{14}$$

where the exact AGP is replaced with an approximation $\chi_\lambda$ in Eq. (12). Minimisation of the action is equivalent to minimising Eq. (11), and so if the operator basis is complete then $\chi_\lambda$

will be equivalent to the exact AGP. In any system with a finite Hilbert space there will exist a finite set of operators which span the space and are able to represent the exact AGP.

To give a physical interpretation of what minimising Eq. (11) means, we define

$$F(\chi_\lambda) = \frac{1}{\dim(\chi_\lambda)} \mathrm{Tr}\left([H(\lambda), G(\chi_\lambda)]^2\right). \tag{15}$$

This quantity $F(\chi_\lambda)$ is the norm of the condition in Eq. (11), and gives a measure of how far away from the exact AGP we are. In Ref. [26] it is shown that this term is proportional to the average rate of change of eigenstate populations. Therefore the smaller this quantity is, the more adiabatic the dynamics are, giving a measure of the success of an approximate expression for the AGP.

### 2.3 Commutator expansion Ansatz

With this knowledge an approximation of the AGP can be computed without resorting to diagonalization of the instantaneous Hamiltonian. However, in general, it is not possible to know the leading set of operators which correspond to a good approximation.

To address this problem, Ref. [48] showed that the AGP can alternatively be expressed as

$$\mathcal{A}_\lambda = \lim_{\epsilon \to 0^+} \int_0^\infty \mathrm{d}t \; \mathrm{e}^{-\epsilon t}(\mathrm{e}^{-iH(\lambda)t}\partial_\lambda H(\lambda)\mathrm{e}^{iH(\lambda)t} - \mathcal{M}_\lambda), \tag{16}$$

with $\mathcal{M}_\lambda$ representing the diagonal terms, which are not relevant in the formulation of the AGP. As such, we focus on the first term in the integral. This can be expanded using the Baker-Campbell-Hausdorff formula to convert between exponentials and commutators. It was shown that only odd terms of the expansion contribute to the off-diagonal terms, giving

$$\mathcal{A}_\lambda = i \sum_{l=1}^\infty \alpha_l \underbrace{\left[H(\lambda), \left[H(\lambda), \ldots [H(\lambda), \partial_\lambda H(\lambda)]\right]\right]}_{2l-1}. \tag{17}$$

This expansion can be truncated to finite order leading to an approximate Ansatz for the AGP. Then, the coefficients, $\alpha_k$, for each order of expansion can be obtained using the variational approach described above. The set of nested commutators is however not necessarily orthogonal, leading to a highly complicated optimisation problem with multiple local minima, making it numerically infeasible in many cases. Here, we seek to address these issues by proposing a new approach for computing the AGP.

## 3 Orthogonal commutator expansion

The approximate methods outlined in Secs. 2.2 and 2.3 have required either *a priori* knowledge of the terms involved in the AGP, or do not necessarily provide stable convergence to the exact result. However, in principle the nested commutator Ansatz of Eq. (17) can represent the exact result, and as such gives all possible operators on which the AGP can have support. This can be used as a basis for the formulation of an algebraic approach to computing the AGP [40]. On a more fundamental level, Eq. (11) already indicates that commutation is a natural language to describe the AGP, allowing for investigation of the underlying structure of this quantity. Here, we look to build upon the previous approaches to develop a numerical method to compute the AGP.

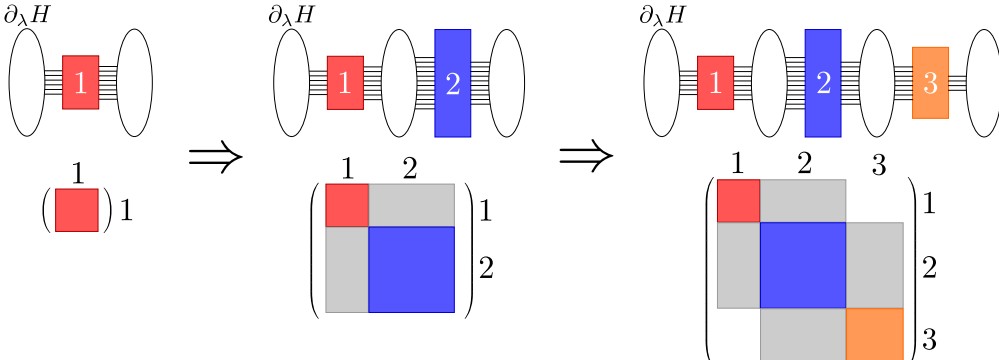

Figure 1: Illustration of the growth of the operators required to compute the AGP. The starting point is $\partial_\lambda H$. We then apply left commutations of the Hamiltonian ($[H,\cdot]$) to compute the next orthogonal set of operators. This process is then iterated as described in the main text. Below each schematic of the required operators we show the structure of the resulting matrix equation which needs to be solved.

The reason for the unstable convergence of the commutator Ansatz, is that the operator created from each expansion is not necessarily orthogonal to all the previous ones. This has been previously addressed using Krylov methods [56], to enforce orthogonality of the operators. We instead suggest decomposing the nested commutators into a chosen basis, and track when each new basis vector appears. The choice of basis is in general unrestricted, similar to the approximate AGP in Eq. (13). However, trace-orthogonal bases provide the simplest representation because the action defined in Eq. (14) is then simply the sum of squares of the coefficients of $G(\chi_\lambda)$.

The operators that can be generated via commutation with the Hamiltonian form a vector space. This space is a Lie-algebra with the commutator as its Lie bracket, which is referred to as the dynamical Lie algebra in the literature [57]. Previous works have shown there are links between the size of the Lie algebra, the controlability of a system [58], and also the degree of quantum chaos in the system [59]. The dynamical Lie algebra consists of all effective operations induced by variation of the Hamiltonian, so it is consistent that the AGP can be defined using it.

The commutator anstatz indicates that the AGP consists only of operators generated via an odd number of commutations. This means there must be a bi-partition between 'odd' operators (AGP) and 'even' operators (the action, $S$). This may appear surprising at first as odd ordered cycles can appear in these Lie algebras, for example $H = \sigma_x + \sigma_y + \lambda\sigma_z$ has a three cycle, so this cannot be bipartite. However this Lie algebra is defined over real coefficients, such that there is distinction between $\sigma_x$ and $i\sigma_x$. As the commutator between Hermitian operators is anti-Hermitian $[A,B]^\dagger = -[A,B]$, this swaps between a Hermitian set of operators and an anti-Hermitian set, which provides the bi-partition. As the AGP of a Hermitian Hamiltonian must also be Hermitian, the odd AGP parts are Hermitian. This change is reflected in Eq. (14) where an extra factor of $i$ is included to ensure the action is real. In general this distinction is more of a technical one, as if an anti-Hermitian operator is used for the AGP, the optimisation coefficient found automatically becomes imaginary to account for this.

For clarity we look at an example of a two site Hamiltonian of the form

$$H = J\sigma_1^z\sigma_2^z + \Delta\left(\sigma_1^x + \sigma_2^x\right) + \lambda\left(\sigma_1^z + \sigma_2^z\right), \tag{18}$$

$$\partial_\lambda H = \sigma_1^z + \sigma_2^z, \tag{19}$$

where $\sigma_i^\gamma$ is the Pauli matrix $\gamma$ acting on site $i$. The first two nested commutations give

$$[H, \partial_\lambda H] = -i\Delta\left(\sigma_1^y + \sigma_2^y\right), \tag{20}$$

$$[H, [H, \partial_\lambda H]] = -J\Delta\left(\sigma_1^x\sigma_2^z + \sigma_1^z\sigma_2^x\right) + \Delta^2\left(\sigma_1^z + \sigma_2^z\right) - \lambda\Delta\left(\sigma_1^x + \sigma_2^x\right). \tag{21}$$

We see already that $\sigma_1^z$ and $\sigma_2^z$ have appeared in both $\partial_\lambda H$ and $[H, [H, \partial_\lambda H]]$. By not keeping track of the new operators we find the set of operators $B_l$ that appear after $l$ nested commutations (taking odd operators to be Hermitian) to be

$$B_0 = \left\{i\sigma_1^z, i\sigma_2^z\right\}, \tag{22}$$

$$B_1 = \left\{\sigma_1^y, \sigma_2^y\right\}, \tag{23}$$

$$B_2 = \left\{i\sigma_1^x\sigma_2^z, i\sigma_1^z\sigma_2^x, i\sigma_1^x, i\sigma_2^x\right\}, \tag{24}$$

$$B_3 = \left\{\sigma_1^z\sigma_2^y, \sigma_1^y\sigma_2^z, \sigma_1^x\sigma_2^y, \sigma_1^y\sigma_2^x\right\}, \tag{25}$$

$$B_4 = \left\{i\sigma_1^z\sigma_2^z, i\sigma_1^x\sigma_2^x, i\sigma_1^y\sigma_2^y\right\}. \tag{26}$$

$$\tag{27}$$

The AGP is then defined over all the sets where $l$ is odd, and we compute the action by commuting onto the sets where $l$ is even.

In general the minimisation processes gives rise to the matrix equation:

$$\mathbf{M}\vec{\alpha} = \vec{\beta}, \tag{28}$$

where the AGP operator coefficients are grouped into the vector $\vec{\alpha}$

$$\mathcal{A}_\lambda = \vec{\alpha} \cdot \hat{\vec{O}} = \sum_p \alpha_p \hat{O}_p. \tag{29}$$

The $(p, k)$ matrix element of $\mathbf{M}$ is computed by summing the structure constants

$$\mathbf{M}_{p,k} = \sum_l c_p^l c_k^l, \tag{30}$$

where

$$c_k^l = -i[H, O_k^{B_{odd}}] O_l^{B_{even}}. \tag{31}$$

These map both odd operators $\hat{O}_p^{B_{odd}}$, $\hat{O}_k^{B_{odd}}$ to any shared even operators $\hat{O}_l^{B_{even}}$. Similarly $\vec{\beta}$ is given by the connection to the initial operators $\partial_\lambda H$

$$\beta_k = -\sum_l c_k^l c_0^l. \tag{32}$$

The zeroth structure constant here is given by

$$c_0^l = \partial_\lambda H \cdot O_l^{B_1}. \tag{33}$$

This form is derived fully in App. A. From this it can be seen there are only non-zero matrix elements between consecutive odd sets, $B_l$. This leads to a block tridiagonal form of the matrix equation which can be efficiently solved [60].

The approach is shown diagrammatically in Fig. 1, where ovals and rectangles represent the even and odd sets of operators, $B_l$, respectively. The lines represent connections, which arise from commutation, between the operators in consecutive sets. In this figure we see how, starting from the zeroth set, $B_0$, of operators which appear in $\partial_\lambda H$, we apply commutation with the Hamiltonian to compute the next set of operators. The first of these sets, labelled 1

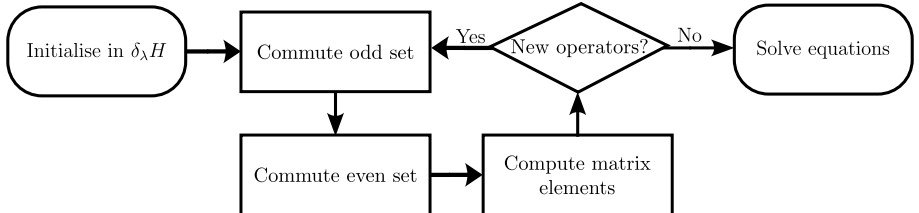

Figure 2: Flow chart of the algorithm. Commutation is repeatedly used to both generate the next operators, and the corresponding matrix equation. Additionally the process can be stopped at the "New operators?" to give a truncated result if the exact result is numerically infeasible. (See App. B for further details.)

in the red rectangle, refers to the first set of AGP operators ($B_1$). Left commutation is then applied to this set giving the blank oval which represents the operators that define the action. By optimising the action, we end up with a linear set of equations dependent on set one. Then repeat this process for the second set, computing the matrix elements from shared operators within the oval set between sets one and two. This then repeats for sets two and three, noting there are no direct connections between sets three and one, so the equations cannot couple these sets. As such in matrix form, we only get non-zero matrix elements between consecutive sets (one-two, two-three). This leads to a block tridiagonal form of the matrix, as shown in the lower section of Fig. 1.

## 3.1 Numerical implementation

We now show how it is possible to implement this approach numerically. The main steps are as follows:

1. Choose a trace-orthogonal basis, where the commutation relations are know, to represent the operators.

2. Start with $\partial_\lambda H$ as the zeroth set.

3. Commute each element with the Hamiltonian, creating the next odd set.

4. Commute again each element with the Hamiltonian, creating the next even set.

5. Compute the structure constants, and build the matrix elements from the shared connections.

6. Repeat steps 3–5 until no new operators are made, or truncation limit is reached.

7. Solve the resulting matrix equation.

These steps are shown in Fig. 2. App. B gives full details of our numerical implementation, alongside a more detailed flow chart.

In many models, the size of the basis for the AGP and, therefore, of **A** grows exponentially with the system size. As a result, controlled approximations are required to keep the problem tractable. This can be done by limiting the depth of the nested commutations (i.e. the value of $l$ above) which are computed, giving a truncated form of the AGP similar to the local counter diabatic approach. The truncation depth can be varied allowing for convergence towards the exact AGP, which is guaranteed when the all basis operators are included. Another approach is to note that there are cases where many different $\alpha_k$'s are exactly equal or very similar to each other. We can then approximate them by grouping these operators together and only calculating a single value. An explicit example of this is given in Sec. 4.4, where we consider a

single local defect in an otherwise symmetric model. By using a combination of these methods truncation and grouping, we have an approach that can efficiently compute the approximate AGP to a given accuracy, which goes beyond the local counter diabatic approach utilised mostly to date, and by increasing this we can guarantee convergence to the exact AGP. In Sec. 4 we will see that this expansion further provides physical insights into the problem due to the structure of the operators appearing.

## 3.2   Overview of method

Our approach leads to a useful pictorial form for representing the AGP. The structure in Fig. 1 shows this, giving a graphical description of how operators are linked to each other via commutation. The structure of this graph can lead to interesting insights into the structure of the AGP. One case where this insight is useful, is explaining the surprising benefit of adding operators from sets at even orders of $l$ into the Hamiltonian leading to improvements in control procedures such as COLD [61]. The effect of adding such operators, changes the ordering and increases the connectivity of operators in the graph, which allows more operators to be controlled, without increasing the size of the dynamical Lie-algebra.

The basis used to generate the Lie-algebra can be freely chosen, although the algebra is much simpler if it is trace orthogonal. This then allows the choice of a physical basis relevant to a particular experimental situation. This makes it straightforward to understand what operators need to be implemented and allows understanding of the approximations which must be made given a set of hardware limitations. Another benefit is that operators can be completely disconnected, leading to the formation of islands of operators (for an example of this see Sec. 4.1). This ensures only the necessary operators in the basis are used to represent the AGP.

There are also numerical advantages in solving the resulting matrix equation compared to previous algebraic approaches [40], and with clear ways of truncating the problem for large Hilbert spaces. It would be very interesting to see how the approach compares to methods such as the Krylov approach [56].

We have created an open source package, mAGPy [62] which implements the methods outlined in this paper. Currently the focus is on systems of coupled spin-1/2 particles, using a binary symplectic form to efficiently store the operators. All the results of this paper are as such reproducible using the package.

## 4   Example: Ising graphs

Quantum annealing has been developed to solve combinatorial optimisation problems with a wide variety of potential applications [32,49,63,64]. Such optimisation problems are often recast into finding the ground state of an Ising-type Hamiltonian [50,65–67] which is normally obtained by adiabatically modifying a Hamiltonian with a trivial ground state towards the Ising model which encodes the optimisation problem. This encoding often requires the couplings between spins in the model to be turned on or off, and is not limited to the crystalline geometries studied in condensed matter physics. This approach has been pursued in a range of experimental setups, including superconducting qubits [68–70], trapped ions [71–73], and Rydberg atoms [51,74]. This means that a natural example of the application of our orthogonal commutator expansion approach is to calculate the AGP for generic transverse field Ising models on various graph geometries. This is inspired by the recent advances in encoding optimisation problems on Rydberg atom hardware [51], though our general approach and the findings described in this example are platform agnostic.

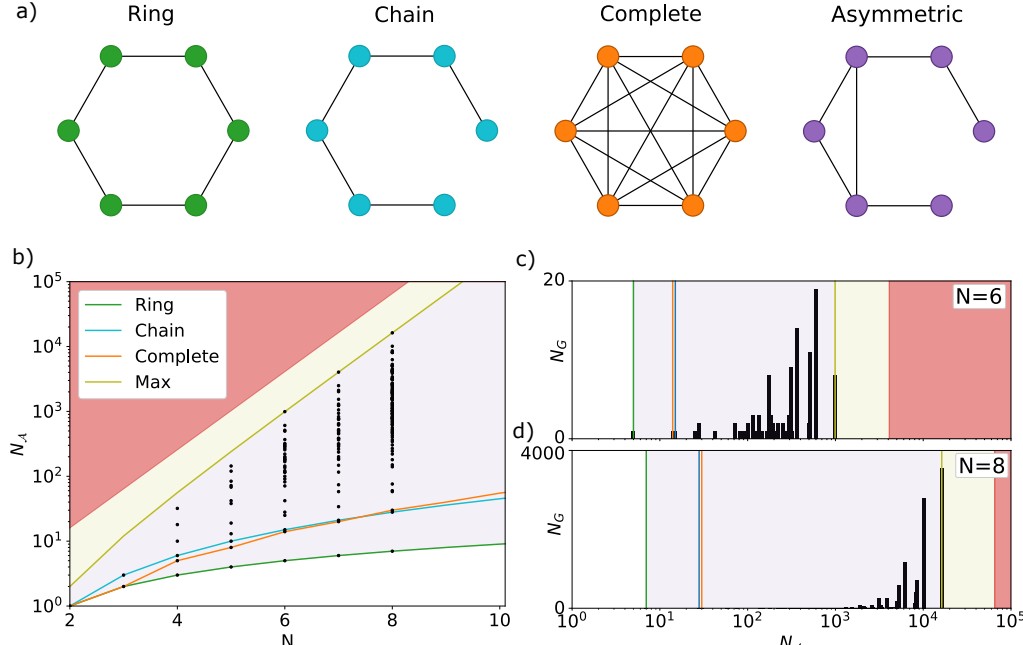

Figure 3: a) Diagrams of three special case graphs, and an example asymmetric graph which has the maximum possible number of coefficients for $N = 6$. b) The scaling of the number of non-zero unique coefficients ($N_{\mathcal{A}}$) required to compute the exact AGP vs. graph size. Black dots represent values for specific graphs, the coloured lines represent the three special cases and the maximum. The purple region shows the possible values of $N_{\mathcal{A}}$, yellow shows the region between the maximum possible for the Ising model studied here and the full size of operator space, and red region shows values large than the operator space. c) For all $N = 6$ graphs, the number at each value of $N_{\mathcal{A}}$. d) Same as c) but with $N = 8$.

The transverse field Ising model on a particular graph is given by the Hamiltonian

$$H(\lambda) = -J \sum_{(i,j)} \sigma_i^z \sigma_j^z + \lambda \sum_i^N \sigma_i^x \,, \tag{34}$$

where $\sigma^x$ and $\sigma^z$ are the usual Pauli matrices, $J$ is the coupling constant which can be positive or negative giving a ferromagnetic or antiferromagnetic interaction, $N$ is the number of sites or vertices in the graph, and $(i, j)$ are the indexes of sites connected by an edge on a given graph. We set $J = 1$ enforcing a ferromagnetic interaction, and use this as our unit of energy. The control parameter $\lambda$ is the strength of the external transverse field. When the graph is that of a chain then the Hamiltonian, Eq. (34), gives the 1D transverse field Ising model which, for $N \to \infty$, has a phase transition at $\lambda = \pm J$. To test the approach we have developed, we will consider the impact of the geometry of a given graph on the adiabaticity of dynamical paths in the Ising model. Examples of the types of graphs we will consider are shown in Fig. 3(a).

Using the orthogonal commutator expansion, we can produce a complete basis for any graph, by commuting and grouping operators as described in Sec. 3. Details of our specific numerical implementation are given in App. B. This importantly leads to two different factors which affect the size of the basis; symmetries and islands. The effect symmetries have on limiting the size of the basis is clear. The automorphism group on the graph tells us which sites can be treated equally, meaning operators acting on equivalent sites can be swapped without any effect. For example, in a two-qubit model, $\sigma_1^y \sigma_2^z$ and $\sigma_1^z \sigma_2^y$ can be grouped together as

the sites are equal. Secondly, as terms are coupled via commutation with the Hamiltonian, it is possible for islands to appear where sections are completely disconnected. If we initialise with $\partial_\lambda H$ in one of these islands, then there is no escape resulting in a greatly reduced number of terms compared to the general case. We will show an example of this in Sec. 4.1 for the ring graph. Together these two effects can allow for the AGP to be obtained via orthogonal commutator expansion for certain graphs in large systems.

We first consider the maximum number, $N_{\mathcal{A}}^{\max}$, of operators which could be required to form the exact AGP for the Ising model on an arbitrary graph. This can generally be found from how many orthogonal operators are generated by the commutation relation in Eq. (17). The form of Ising model we have written in Eq. (34) has a $\mathcal{Z}_2$ symmetry, which means there must be an even total number of $\sigma^y$ and $\sigma^z$ operators, as individually these break the $\mathcal{Z}_2$ symmetry. We can strengthen this limitation further by using the fact we have a real Hamiltonian, so the AGP must be fully imaginary [26]. As only $\sigma^y$ is imaginary, it is necessary and sufficient to impose an odd number of $\sigma^y$, leading to an odd number of $\sigma^z$ as well such that the total is even. By counting all permutations of odd $\sigma^y$ and $\sigma^z$, we get

$$N_{\mathcal{A}}^{\max} = \sum_{n_y}^{N} \sum_{n_z}^{N-n_y} \frac{N!}{n_y! n_z! (N - n_y - n_z)!} 2^{N - n_y - n_z} \,, \tag{35}$$

where $n_y$, $n_z$ are the, strictly odd, number of $\sigma^y$ and $\sigma^z$ operators. This can be simplified to

$$N_{\mathcal{A}}^{\max} = 2^{N-1}(2^{N-1} - 1)\,. \tag{36}$$

Asymptotically, this gives $4^{N-1}$ operators meaning there is a reduction by a factor of 4 from the full operator space which is of size $4^N$. However, computing the exact AGP is still a fundamentally exponentially scaling problem, unless there are symmetries or islands to exploit.

In Fig. 3(b) we show the value of $N_{\mathcal{A}}$ for all non-isomorphic graphs up to size $N = 8$. Note, this calculation only depends on the particular graph chosen and does not involve obtaining the coefficients of the operators for the AGP, which could be zero, and is independent of $\lambda$. The value for each graph is shown by a black dot. The four lines show results for cases where the exact scaling is known (see Secs. 4.1–4.3). The value of $N_{\mathcal{A}}$ lies within the purple region which is bounded by the maximum case of Eq. (36), and a lower bound given by the most symmetric graph structure, the ring graph. In particular, we note that for $N \geq 6$ there are points that lie on this upper bound which are asymmetric, and have enough connectivity to avoid islands limiting the number of operators required (see Sec. 4.4).

Figure 3(b) only shows the values which $N_{\mathcal{A}}$ can take, but gives no indication of the amount of graphs in which each value occurs. The amounts are shown in more detail in Fig. 3(c)–(d) for graphs of size $N = 6$ and $N = 8$. We find that as the size of the graph increases the fraction of all graphs which have the maximum possible value of $N_{\mathcal{A}}$ increases. This is because the proportion of asymmetric graphs increases along with the graph size, a known result from graph theory [75]. For large random graphs there is very little suppression in the number of operators required in the AGP from symmetry, and only islands can affect the results. This means it is unfeasible to compute the exact AGP due to the exponential scaling of the number of operators required. In such a scenario, information on the adiabaticity of a procedure could be obtained by taking an appropriate truncation of the orthogonal commutator expansion or by only permitting certain operators to be retained, e.g., omitting long-range operators. The best form of approximate AGP will vary depending on the problem and this is better studied by applying the orthogonal commutator expansion approach to specific settings. Below we consider in detail the special graphs of the ring, chain, and complete graph, before discussing the more general case of asymmetric graphs.

## 4.1 Ring graph

The simplest graph on which to compute the AGP for the Ising Hamiltonian is the ring, where the connected edges are those with $(i, j) = (i, i + 1)$ and periodic boundary conditions are applied with $N + 1 \equiv 1$ as shown in Fig. 3(a). This model has been well studied in the literature [76,77] and serves as a good first step for applying the method. For extra mathematical details of the content presented in this subsection see App. C. It can be shown that all operators which appear in the AGP of a one-dimensional system have only one $\sigma^y$ and $\sigma^z$ operator, with $\sigma^x$ operators between them. As such, the form of these operators can be written in the form $B_k^i = \sigma_i^y \sigma_{i+1}^x \ldots \sigma_{i+k}^x \sigma_j^z$, where $j - i - 1 = k$. In addition, the ring has an $N$-fold discrete rotation symmetry and reflection symmetry around any point, meaning that all oeprators are equivalent for a given length $k$. This means that the maximum number of operators to form the exact AGP must be given by

$$N_{\mathcal{A}}^{\text{ring}} = (N - 1), \tag{37}$$

i.e., it is linear in the system size. Putting this together the AGP is then given by

$$\mathcal{A}_\lambda = \sum_k \alpha_k \sum_i B_k^i. \tag{38}$$

In this simple case, finding the values of $\alpha_k$ and hence the AGP turns into solving a symmetric, idempotent, tridiagonal matrix equation with the form of Eq. (28). This can either be solved efficiently numerically but also has a closed-form solution

$$\alpha_k = \frac{\lambda^{k-1}}{8} \frac{\lambda^{2(N-k)} - 1}{\lambda^{2N} - 1}. \tag{39}$$

Note, the coefficients are given for the $k$th order of the commutator expansion in Eq. (17). As we have an analytical form for the coefficients, we can directly take the infinite system size limit, $N \to \infty$, to obtain

$$\lim_{N \to \infty} \alpha_k = \begin{cases} (-1)^k \frac{\lambda^{k-1}}{8}, & 0 \le \lambda \le 1, \\ (-1)^k \frac{\lambda^{-k-1}}{8}, & \lambda > 1. \end{cases} \tag{40}$$

We plot the first 100 $\alpha_k$ for the infinite system in Fig. 4(a). In this figure, the even and odd length terms are indicated by blue and red respectively, with the color fading towards white as the size ($k$) is increased. This shows that $\alpha_k$ decreases in magnitude as $k$ increases, with it flipping sign between positive and negative for even and odd $k$ respectively. This means that the more local terms such as $\sigma_i^y \sigma_{i+1}^z$ are the most important to counteract. However, at the point of the phase transition every $\alpha_k$ is equal and the AGP norm diverges. We can directly compute the norm, defined in Eq. (7), as

$$\lim_{N \to \infty} \frac{\|\mathcal{A}_\lambda\|^2}{N} = \begin{cases} \frac{1}{32} \frac{1}{1 - \lambda^2}, & 0 \le \lambda \le 1, \\ \frac{1}{32\lambda^2} \frac{1}{\lambda^2 - 1}, & \lambda > 1. \end{cases} \tag{41}$$

Note that this is divided by a factor of $N$; even away from the critical point the multiplicity of each term ensures that the norm is $\propto N$. This result shows that far above ($\lambda \gg 1$) the critical point the AGP is almost zero but far below ($\lambda \ll 1$) the AGP norm converges to a constant, as shown in Fig. 4(d). This occurs because the eigenstates of the two body interaction term $\sigma_i^z \sigma_{i+1}^z$ are more affected by the perturbations of the small anisotropy of the $\sigma_i^x$ term, as the spins are strongly correlated. Whereas far above the critical point all the spins are independent from each other, so perturbations due to the coupling are infinitesimally small. However, the norm only asymptotically reaches zero with an inverse power law. The AGP norm is exactly zero for $\lambda = 0$ and in the limit $\lambda \to \infty$ as these relate to Hamiltonians: $H(\lambda = 0) = \sum_i -J \sigma_i^z \sigma_{i+1}^z$ and $\lim_{\lambda \to \infty} \frac{H(\lambda)}{\lambda} = \sum_i \sigma_i^x$ which only have one term, giving no AGP as it commutes with its derivative.

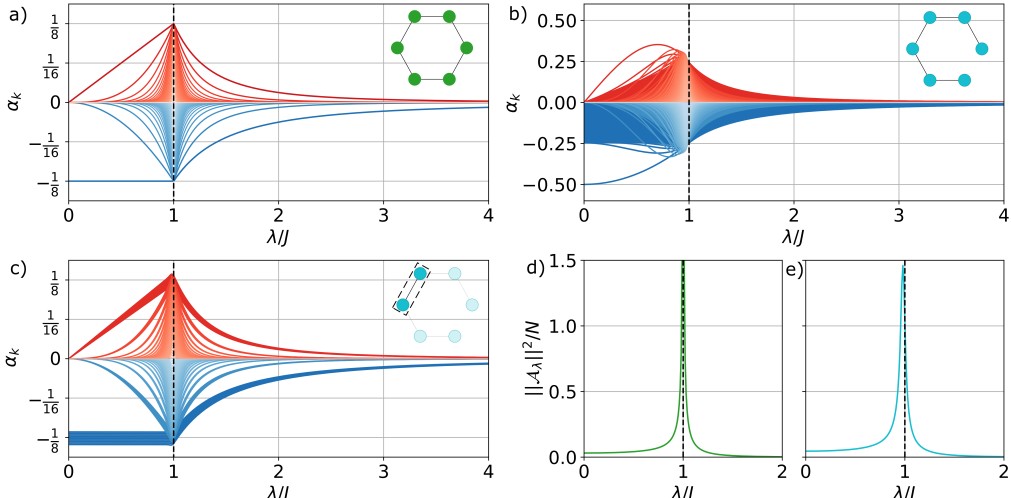

Figure 4: a) Coefficients of unique terms in the thermodynamic limits of the ring graph. Blue and red differentiate between operators with even and odd length respectively, the color fades as the length of the operator increases. The critical point is indicated at $\lambda = 1$ b) Coefficients of unique terms in the $N = 100$ chain graph, with the same colouring as before. Panel c) shows what happens when the terms are limited to those which only have support on the middle 10 sites for the $N = 100$ chain, same colouring as before, d) Norm of the AGP for the ring graph and e) AGP norm of the $N = 100$ chain graph.

## 4.2 Chain graph

The next case we consider is the chain graph. This is identical to the ring above but we break one link giving a chain with open boundary conditions. The chain graph no longer has the rotational symmetry of the ring graph and only has a reflection symmetry around its center. This only provides a reduction of half from the symmetry, however the island effect for 1D graphs still reduce the operator space. As stated before for the ring graph, we have terms of the form $\sigma_i^y \sigma_{i+1}^x \ldots \sigma_{i+k}^x \sigma_j^z$, which if we count the possible combinations and take into account the factor of a half from symmetry, we get a quadratic scaling of

$$N_{\mathcal{A}}^{chain} = \frac{N}{2}(N-1). \tag{42}$$

This allows for efficient computation of the AGP coefficients up to large system sizes. We show results in Fig. 4(b) for a chain of $N = 100$ spins. We again find that there is a divergence in these values close to the location of the phase transition as expected, and the scaling of the divergence is the same as that seen in the ring graph. The terms which have only two operators are, on average, the largest. They are the ones which start at a non-zero value at $\lambda = 0$, but the terms that hit the largest maximum values are terms with $\sigma^y$ on the ends. This means that, there are different terms that are more important at different values of $\lambda$.

In Fig. 4(c), we limit the terms we show to those with support only on the middle 10 sites. As expected, far away from the boundaries, the chain graph behaves very similarly to the ring graph. This can be seen by comparing the results to those in Fig. 4(a). As such it should be possible to write an approximation for the chain in which we treat a number of terms around the boundaries exactly, and then use the analytic results of the ring for the remaining terms. We do not implement this idea here, as we can already reach large enough system sizes to see the emergence of the thermodynamic limit. However, this approach can be used to reduce the scaling and complexity of more difficult problems.

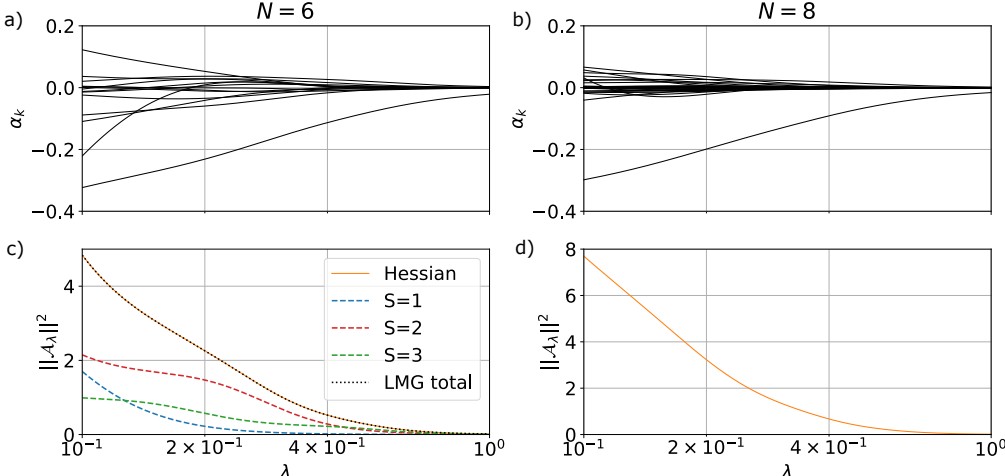

Figure 5: Coefficients of the different operators in the AGP for the complete graph for a) $N = 6$ and b) $N = 8$. The dominant term in both cases is $\sigma^y \sigma^z$. Panel c) shows the AGP norm for $N = 6$ graph. Along with this are the norms obtained for relevant LMG models as described in the main text. Panel d) gives the AGP norm for the $N = 8$ graph.

## 4.3 Complete (all-to-all) graph

We next go on to study the complete graph which has all possible connections present, this gives rise to an Ising model with all-to-all connectivity. In this case the AGP can contain all possible terms that satisfy the symmetries of the Hamiltonian. Whilst this is a very large number of operators, the complete graph is *vertex-transitive*, meaning that all sites are equal and hence all permutations of a term can be grouped together. This leads to an effective scaling of

$$N_{\mathcal{A}}^{\text{complete}} = \begin{cases} N(N + 1)\left(\frac{N+1}{2} - \frac{2N+1}{3}\right), & N \text{ even}, \\ N(N - 1)\left(\frac{N+1}{2} - \frac{2N-1}{3}\right), & N \text{ odd}. \end{cases} \tag{43}$$

It is important to note the difference between $N$ being odd and even, which can be best explained by transforming to a collective spin model. The complete graph conserves the total spin and so can be viewed as a set of nested collective spin models which do not couple to each other. These models are described by the Lipkin-Meshkov-Glick (LMG) Hamiltonian [78]

$$H_{\text{complete}} = \sum_{n}^{N/2} C_n^N H_{\text{LMG}}^n, \tag{44}$$

where the individual LMG Hamiltonians are given by

$$H_{\text{LMG}}^n = \frac{J}{N}\left(S_z^n\right)^2 + \lambda S_x^n. \tag{45}$$

We have defined the collective spin operators $S_x^n$, $S_z^n$ which have a total spin $S = n$. The multiplicity factor, $C_n^N$, counts the unique ways of combing $N$ spin half particles to get total spin $S = n$. This multiplicity corrects for the inherent difference in Hilbert space dimension of the complete graph ($D \approx 2^N$) and the individual LMG models ($D \approx N$). If $N$ is even the sum runs over integers from 0 to $N/2$ while if $N$ is odd it runs over half integers in the same range. With this we can see that when $N$ is even all the spin sectors which contribute are bosonic. However, when $N$ is odd, these are fermionic. This fundamental change leads to a difference

in the scaling and the form of the AGP between the two cases. As such it makes sense when comparing between different values of $N$ to distinguish between when it is odd and even. We choose to show results for even $N$ here, however, we have computed the AGP for odd $N$ and find similar behaviour.

In Fig. 5 we show how the coefficients in the AGP vary with $\lambda$ for different system sizes. We compare results for $N = 6$ in panels a) and c) with $N = 8$ in panels b) and d). These are both even giving bosonic spin sectors for the LMG decomposition. Over the region we show, we see that there is one term that is dominant over the other terms. This line in both plots is for $\sigma^y \sigma^z$, which indicates the importance of terms consisting of only a few operators, similar to the ring and chain graphs. However, for $N = 6$, there is one term with a similar magnitude to this at small values of $\lambda$. This result corresponds to the operator $\sigma^y (\sigma^z)^5$ which has operator content on each site. So whilst in most regions of parameter space, terms with only a few operators are the most important, this is not always true.

In Figs. 5c)–d) we show the AGP norms, which have large peaks near $\lambda = 0$ which arise from a single dominant contribution. For $N = 6$ we show how the contributions to the AGP norm break down into $S = 3, 2, 1$ spin sectors (noting the $S = 0$ sector is ignored as it only has one eigenstate, and so the AGP is always zero). For each of these, we have included the appropriate multiplicity so that the sum of these lines gives the total AGP norm, as can be seen by the exact match between this sum and the result of a full numerical calculation.

It may be surprising that there is no significant peak at the location of the phase transition in this model. This is because we are considering the entire operator space, which has many separate subspaces for each of the different spin sectors. The phase transition occurs in the ground state of the system, and as such is only present in the $S = N/2$ spin sector at $\lambda = 1$ for $N \to \infty$. The other spin sectors, have equivalent 'critical' points at $\lambda = S/N$. These smaller collective spin models have a much larger multiplicity than the $S = N/2$ case, hence they have a large contribution to the norm of the AGP near $\lambda = 0$ which then quickly falls off. Note, that if you are only interested in adiabatically following the ground state of this model, then the Hilbert space can be restricted to that of the $S = N/2$ model as discussed above. This not only gives a significant reduction in the size of the Hilbert space but could also give a significant simplification in the AGP, which would only describe diabatic transitions involving this state, and any counterdiabatic driving that it informs. However, this would require the restriction of the AGP to a specific state, which would require modification of the orthogonal commutator expansion approach which we leave for future work.

We emphasis again the differences between the all-to-all model (complete graph) and the LMG model of just the largest spin sector. The ground state and dynamics within the largest spin sector of these models are identical and hence, often they are considered interchangeable. Here we have seen that for some quantities there are differences. The all-to-all model is equivalent to many independent LMG models, with different total spin quantum numbers combined. We hope this example of computing the AGP norm, and seeing the ground state phase transition being hidden behind these many LMG models, helps clarify why this distinction between models is important.

## 4.4 Asymmetric graphs

We now discuss what is possible when there are no symmetries present and the AGP includes all possible operators. As we stated above, these models have the worst possible scaling as given in Eq. (36). The asymmetry means that all permutations are unique and cannot be grouped, and there is a sufficient degree of connectivity to break apart the one-dimensional sections which are seen in the chain and ring.

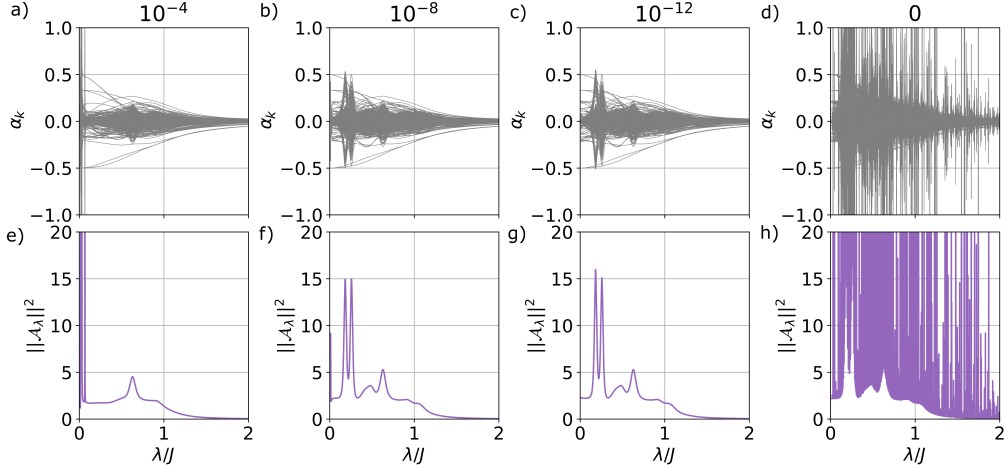

Figure 6: Coefficients, $\alpha_k$, a)–d) and AGP norm e)–h) obtained for the asymmetric graph (as illustrated in Fig. 3(a)). Each column represents a different threshold value for removing entries from the Hessian matrix. The values for these thresholds are indicated at the top of each column.

To examine how our method deals with these asymmetric graphs we focus on one in particular. We take a graph constructed from a chain of length 6 and add an extra connection between sites 2 and 4. This is in the group of graphs that first hit the upper limit shown in Fig. 3(b), and has $N_{\mathcal{A}} = 992$. To be able to compute the AGP corresponding to the Ising model on this graph using previous methods would be extremely difficult. We show in Fig. 6 the results of applying the orthogonal commutator expansion to this graph. In this figure, we show the effect of setting entries in the Hessian matrix smaller in magnitude than a particular threshold to zero. The threshold is decreased across the figure from left to right, increasing the number of entries included. We see that with the smallest possible threshold numerical instabilities arise and cause a divergence in some of the coefficients of the AGP. However, when the threshold is set to intermediate values, as in the middle two columns, the values obtained are stable and give rise to an AGP which accurately follows the ground state of the model. When the threshold is too large this no longer occurs and the choice of threshold starts to affect the calculated AGP.

These results are particularly interesting when simulating the dynamics of the counterdiabatic Hamiltonian on this asymmetric graph. Now, as the AGP norm has large differences between the different threshold values, we may expect the fidelity with the chosen eigenstate to be smaller for thresholds where the AGP is clearly unstable (such as when the threshold is set to numerical tolerance). However, we find that, for following the ground state, the fidelity is approximately $\approx 1 - 10^{-5}$ for all choices of threshold. This indicates that, either the ground state is not affected by the terms that are varying widely, or that the fidelity is not a sensitive enough measurement to pick up on these differences. We also tested following some arbitrary eigenstates and found the same result of fidelity depending very little on threshold value. In addition, we checked for threshold values much larger than those shown here, which gave fidelities that oscillated between 0 and 1 depending on $\dot{\lambda}$. This means that the threshold value does matter in general, just not between the values shown here of $10^{-4}$ to 0. Further study is required to understand the link between fidelity and the accuracy of the AGP in order to have a better understanding the robustness of the counterdiabatic Hamiltonian.

In the stable region, we find that the $\alpha_i$ values are all approximately bounded between $\pm 0.5$. At specific values of $\lambda$ they combine together to give a large norm. These values are related to the specific geometry of the graph in a non-trivial way. However, in certain limits,

there are connections to the calculations and scaling we found for other graphs. If we imagine extending this graph by adding two sites to its ends, the graph would still be asymmetric but for $N = 8$. This process can be repeated until we have an extremely long chain, with a small defect of one extra connection in the center. In this limit, the results would be very similar to those seen for the chain graph. The graph we have constructed is still asymmetric so formally has an exponential scaling of the number of terms which are non-zero rather than quadratic. From this analysis, we see that many of the $\alpha_i$ values should be very similar, and hence approximate solutions could be used to reduce the complexity of the computation. Understanding how to appropriately use this kind of approximation of the AGP is a key fundamental step required for progress towards calculating the AGP for arbitrarily complicated models with no symmetries.

## 5   Conclusions

We have outlined a new approach to calculating diabatic terms for dynamical problems. By utilising a commutator-based approach we obtain the operators which contribute to the AGP which fully describes the diabatic evolution. We have shown that, by starting from this commutator expansion, finding the AGP can be recast into the problem of solving a block-tridiagonal system of equations which are at most quadratic, allowing for efficient approaches to solving for the coefficients of the AGP. An advantage of our approach is that the AGP is given in terms of a physical basis (e.g. the Pauli matrices for spin-1/2) allowing for physical interpretation of results. However, we have found that computing the AGP still scales exponentially in general. This means that approximations are required for large system sizes in the majority of cases. With our approach, three approximation methods are possible: (1) truncating the commutation at a certain order, (2) grouping particular coefficients of terms together and (3) placing restrictions on the operators tracked. Every extra commutation included is guaranteed to improve the result until there are no new terms added and the exact case is reached. Similarly, splitting a group of coefficients into independent terms will either improve the result or give the same result (if the terms are actually connected via symmetry). Due to this, we can tune the approximation until the required convergence is met, or to obtain the best possible approximation of the AGP with a particular numerical cost.

A different approach to overcoming the problems of calculating the AGP is to construct an orthogonal basis using Krylov methods [56]. With this approach, the difficulty in computation is moved to the construction of the basis, as the resulting matrix equation in the chosen Krylov basis is tridiagonal and can be analytically solved. One subtlety of this approach is determining when to stop the generation of basis states, as there is not immediately a clear condition as for the orthogonal commutator expansion. If the correct number of terms are included such that the basis vectors span the entire AGP, then the results of both approaches are equivalent. When approximations are made to the expression for the AGP, either via the truncation of the Krylov basis or commutator expansion, it is not immediately clear under what conditions which approach will be beneficial over the other. This is an area where further study would be interesting.

We have used our approach to study the impact of geometry on the AGP. By studying the transverse field Ising model with different graph connectivities. Analytical results for simple graphs, such as the ring, can be computed. In more complicated graphs, our algorithm can be numerically implemented to compute the matrix equation, which can then be solved. In this setting, we can envisage the AGP being a useful to enforce certain dynamical paths, e.g., in quantum annealing, or to inform how information spreads dynamically through quantum many-body systems. In this way, the AGP is a property worthy of detailed future study in dynamical problems beyond its application in the field of quantum optimal control.

# Acknowledgments

The authors acknowledge K. Damezin, P. Claeys, S. Morawetz, and A. Polkovnikov for useful discussions. The data for this manuscript is available at Ref. [79].

**Funding information** E.D.C.L acknowledges funding from EPSRC grant EP/T517938/1. C.W.D. acknowledges funding from EPSRC grant EP/Y005058/1.

# A   Derivation of Hessian form

In this Appendix we present a derivation of Eqn. (28) from the main text. To calculate the values of $\alpha_k(\lambda)$ we need to minimise the action $S_\lambda$ given by:

$$S_\lambda = \text{tr}\big[G_\lambda^2\big], \tag{A.1}$$

$$G_\lambda = \partial_\lambda H - i\,[H, \mathcal{A}_\lambda]\,. \tag{A.2}$$

It is useful to note here that $G_\lambda^2$ can be used in place of $G_\lambda G_\lambda^\dagger$ as $G$ is Hermitian. As such we are required to compute the derivative $\nabla S_\lambda = 0$ with respect to all the $\alpha_k(\lambda)$. We denote the bases of $\mathcal{A}_\lambda$ and $G_\lambda$ as $\{O^A\}$ and $\{O^G\}$ respectively. As we have mentioned in the main text, the odd operator sets in Fig. 1 represent $\mathcal{A}_\lambda$, meaning the expression $[H, \mathcal{A}_\lambda]$ will map onto all even operator sets, which represent $G_\lambda$. Hence left commutation with $H$ maps $\{O^A\}$ onto $\{O^G\}$. Let us express this in the following way

$$\begin{aligned}
G_\lambda &= \partial_\lambda H - i\left[H, \sum_k \alpha_k O_k^A\right] \\
&= \partial_\lambda H + \sum_l \left(\sum_k c_k^l \alpha_k\right) O_l^G,
\end{aligned} \tag{A.3}$$

where $c_k^l = -i[H, O_k^A] \cdot O_l^G$. Here $\partial_\lambda H$ is also defined on $\{O^G\}$, so we can count from $k = 1$ and include $k = 0$ as a constant term not dependent on $\alpha_k$:

$$G_\lambda = \sum_l \left(c_0^l + \sum_k c_k^l \alpha_k\right) O_l^G, \tag{A.4}$$

where $c_0^l = \partial_\lambda H \cdot O_l^G$. We then use the fact that a Hilbert space always has a trace orthonormal basis [80]. For spin systems, a basis that satisfies this property is the generalised Gell-Mann matrices as they are traceless, and are generalisations of the Pauli matrices for spin 1/2 to larger spins. These provide a basis for the operator groups which can extended to the full system by using tensor products of each of these operator group basis operators. With this property, we get:

$$\begin{aligned}
S_\lambda &= \sum_{m,l} \left(c_0^m + \sum_n c_n^m \alpha_n\right)\left(c_0^l + \sum_k c_k^l \alpha_k\right) \text{tr}\big[O_l^G O_m^G\big] \\
&= \sum_{m,l} \left(c_0^m + \sum_n c_n^m \alpha_n\right)\left(c_0^l + \sum_k c_k^l \alpha_k\right) S_0 \delta_{l,m}, \\
S_\lambda/S_0 &= \sum_l \left(c_0^l + \sum_n c_n^l \alpha_n\right)\left(c_0^l + \sum_k c_k^l \alpha_k\right),
\end{aligned} \tag{A.5}$$

where $S_0 = \mathrm{tr}\big[O_l^G O_l^G\big]$ is some constant factor which we assume to be the same for all $l$, for example for spin $1/2$ this is $S_0 = 2^N$. This expression for $S_\lambda$ shows that this is a quadratic form in $\alpha_k$, meaning there is only one stationary point of $S_\lambda$. We can also further state that this must be a minimum, as the AGP exists for all $H$, as such this value must have a lower bound. If this was not the case then the AGP would be ill-defined, as there is always a way to further reduce $S_\lambda$ with no asymptotic solution for large $\alpha_k$ possible because of the quadratic form. Now we can compute the partial derivatives with respect to $\alpha_p$:

$$\frac{\partial}{\partial \alpha_p} S_\lambda / S_0 = \sum_l c_n^l \delta_{n,p} \left( c_0^l + \sum_k c_k^l \alpha_k \right) + \left( c_0^l + \sum_n c_n^l \alpha_n \right) c_k^l \delta_{k,p}$$

$$= \sum_l 2 c_p^l \left( c_0^l + \sum_k c_k^l \alpha_k \right), \tag{A.6}$$

where we have renamed the dummy index $n$ to $k$ to consolidate into one term and get a factor of 2. We can now apply the condition $\frac{\partial}{\partial \alpha_p} S_\lambda = 0$ to give:

$$\sum_k \sum_l c_p^l c_k^l \alpha_k = -\sum_l c_p^l c_0^l. \tag{A.7}$$

If we then combine all the different partial derivatives (as all must be 0), we can write a matrix equation:

$$\sum_{p,k} \sum_l c_p^l c_k^l \alpha_k = -\sum_p \sum_l c_p^l c_0^l,$$

$$\mathbf{M} \vec{\alpha} = \vec{\beta}, \tag{A.8}$$

where:

$$\mathbf{M}_{(p,k)} = \sum_l c_p^l c_k^l, \tag{A.9}$$

$$\vec{\alpha}_{(k)} = \alpha_k, \tag{A.10}$$

$$\vec{\beta}_{(k)} = -\sum_l c_k^l c_0^l. \tag{A.11}$$

This is the result used in the main text.

# B  Numerical implementation

In this appendix we describe how we numerically implement the procedure described in Sec. 3 of the main text. To achieve this the operators which we keep to form part of the AGP and the action must be represented in an efficient form for the computer to use. The full matrix form is not required to evaluate the commutators. We must keep enough information be able to work out the sign and form of a new term after application of a commutation with $H$. The weight for each term will be accounted for separately when solving for the $\alpha_k$ values. The simplest approach to achieve this is to use strings to represent the operators, for example $\sigma_1^x \sigma_2^z \mathbb{I}_3 \sigma_4^y$ can be written as 'xzIy' for a model consisting of four spin-1/2 particles. Commutation can be implemented via dictionaries. For example {'y':(1, 'z'), 'z':(-1, 'y')} gives the action of $[x, ]$ since $[x, y] = z$ and $[x, z] = -y$. Looping over the sites applying the commutation at each point can then generate the required result. This approach is quite straightforward, however the size of the dictionaries can be very large if the Hamiltonian has terms with a large number

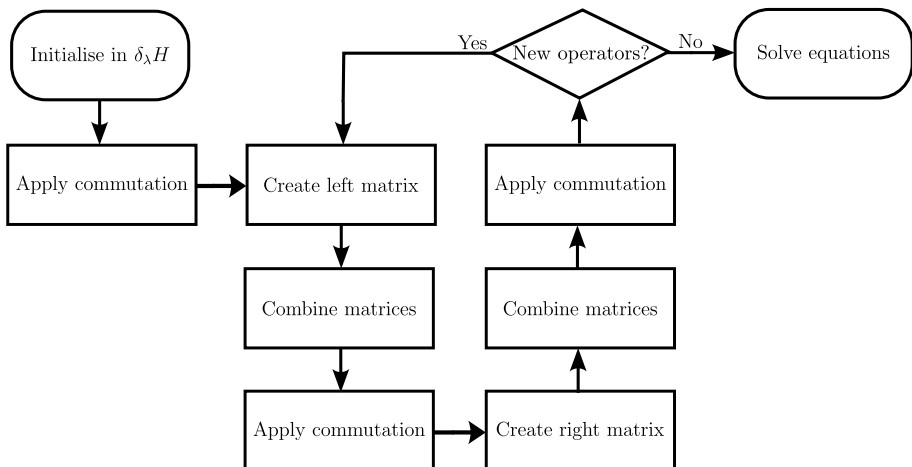

Figure 7: Detailed flow chart of the algorithm. Commutation is repeatedly used to both generate the next terms, and the corresponding matrix equation. Additionally the process can be stopped at the "New Operators?" to give a truncated result if the exact result is numerically infeasible.

of operators as we need to account for all possible combinations, and strings are not the most memory efficient way of representing small operator spaces. For spin-1/2 systems, the most computationally efficient way to encode the operators would be the Pauli symplectic form that was developed by the quantum error correction community [81].

With a given implementation of the encoding of operators and their commutations relations the steps of the algorithm can be followed, as seen in Fig. 7 as a flow chart. We now describe each step in more detail:

Initialise in $\partial_\lambda H$   First initialise the first even operator group (zero) to the derivative of the Hamiltonian. If there are known symmetries, then operators can be grouped by this. Note that only one operator from each symmetry group is required to be propagated, as they have the same structure as each other. However it is important to track the multiplicity of each symmetry group in this case, as this will affect the coefficients of the Hessian matrix built later.

Apply commutation   Apply a commutator of the form $[H,.]$ to the current operator group, and collect the new operators for the next operator group. Note it is important to store the operators separately for odd and even operator groups, if the basis sets have overlap. This ensures that new operators can be checked at later steps. Similarly to initialising, group operators by symmetries if these are known.

Create left matrix   It is important to track the connections between operators, so that the Hessian matrix can be built. To track the connections between operators, we create a matrix for each operator. The columns of the matrix represent what operator you originated from, and rows represent operator commuted onto. For each connection add the sign of the commutation, taking into account the multiplicity of symmetry groups if grouping operators by symmetries. As such if the matrix element is zero or nonzero then there is no connection or a connection respectively. Note in addition we must multiply this matrix by minus one, as the Hessian is

defined from the connections from all odd operator groups to even operator groups, and in this step we are going from even to odd.

Create right matrix   This step is almost identical to creating the left matrix, however the columns and rows are flipped. So now the rows of the matrix represent what operator you originated from, and columns represent operator commuted onto. Also we do not need to multiply by minus one, as we are tracking odd to even operator groups this step.

Combine matrices   At this step combine the latest left and right matrices together into part of the final matrix equation. This is done by computing:

$$\begin{pmatrix} M_R M_R^T & M_R M_L^T \\ M_L M_R^T & M_L M_L^T \end{pmatrix},$$

for each of the combinations of $M_R, M_L$ for the different operators. This matrix can then be added to the associated blocks in the full hessian matrix (see Fig. 1 for a visual guide). Note the first and last time this is done, there will only be one new left or right matrix, so just take the square part $M_L M_L^T$ or $M_R M_R^T$ and place in the block of the associated odd operator group.

New operators?   Check whether the previous commutation generated any new operators, if not then the full set has now been spanned for both odd and even numbers of commutations.

Solve equations   Now the full Hessian matrix equation has been generated, which can then be solved for given values of the variable coefficients using routines from standard linear algebra packages.

The algorithm as described above has scaling: $\mathcal{O}(N_H(N_{odd} + N_{even}))$ where $N_H$ is the number of operators in the Hamiltonian, and $N_{odd}$, $N_{even}$ are the number of operators for odd, even commutations. Implementing the symmetries of the Hamiltonian can greatly reduce the number of operators that need to be considered. Note, that unlike other numerical algorithms this only needs to be computed once for a Hamiltonian, as it represents all the operators symbolically. Once the matrix equation is generated it will then need to be solved for specific parameters numerically or solved analytically. Due to the structure of the matrix, banded methods such as LU decomposition can be very efficient.

## C  Details of calculations for the ring graph

In this Appendix, we give more details of the calculation of the AGP for the ring graph described in Sec. 4.1. This graph is special as we can get a full analytical solution, allowing a thermodynamic limit to be taken. In this graph, each spin is connected to its neighbours, with the first and last spins also being connected. It is important to note that $N = 2$ is a special case, and we focus here on $N \geq 3$ where the ring is clearly defined. This gives an adjacency

matrix of the form:

$$
\begin{pmatrix}
0 & 1 & 0 & \dots & \dots & 0 & 1 \\
1 & 0 & 1 & \ddots & \ddots & \ddots & 0 \\
0 & 1 & 0 & \ddots & \ddots & \ddots & \vdots \\
\vdots & \ddots & \ddots & \ddots & \ddots & \ddots & \vdots \\
\vdots & \ddots & \ddots & \ddots & 0 & 1 & 0 \\
0 & \ddots & \ddots & \ddots & 1 & 0 & 1 \\
1 & 0 & \dots & \dots & 0 & 1 & 0
\end{pmatrix}.
\tag{C.1}
$$

We can define the symmetries of the graph with two generators, a clockwise rotation of all the spins, and a mirror around the central point. In addition, the one-dimensional nature of the ring, the terms become limited to having $\sigma^x$ terms in the bulk, with $\sigma^y$ and $\sigma^z$ on either end. This used in addition to the symmetries means we can group together terms with an equal number of terms. The easiest way to show the operators are of this form is to define the connections for each operator. We can represent these operator connections like so:

$$
\begin{array}{c}
x \qquad\qquad yy \qquad\qquad yx^{l-1}y \qquad\qquad yx^{l}y \qquad\qquad yx^{N-3}y \qquad\qquad yx^{N-2}y \\
\nwarrow 4iJ \qquad \nearrow -4i\lambda \qquad\qquad \nwarrow 4iJ \qquad \nearrow -4i\lambda \qquad\qquad \nwarrow 4iJ \qquad \nearrow -4i\lambda \\
yz \qquad\qquad\qquad\qquad yx^{l}z \qquad\qquad\qquad\qquad yx^{N-2}z \\
4i\lambda \swarrow \qquad \searrow 4iJ \qquad\quad 4i\lambda \swarrow \qquad \searrow 4iJ \qquad\quad 4i\lambda \swarrow \qquad \searrow 4iJ \\
zz \qquad\qquad zxz \qquad zx^{l}z \qquad\qquad zx^{l+1}z \qquad zx^{N-2}z \qquad\qquad x^{N-1}
\end{array}
$$

Here we see that the starting point $\partial_\lambda H = \sum_i^N \sigma_i^x$ is included, and each only has connections to $\sigma_y \sigma_z$ operators. Otherwise, all connections are explored, and there is no way to break away to different operators. Note, that the coefficient of the connections is doubled to four in every case because there are always two different operators (that have been grouped together) that map to the same thing. With these connections, we can write out the matrix equation for the graph as

$$
\begin{pmatrix}
\lambda^2 + J^2 & -J\lambda & 0 & & & & \\
-J\lambda & \lambda^2 + J^2 & -J\lambda & & & & \\
0 & -J\lambda & \lambda^2 + J^2 & & & & \\
& & & \ddots & \ddots & \ddots & \\
& & & \lambda^2 + J^2 & -J\lambda & 0 \\
& & & -J\lambda & \lambda^2 + J^2 & -J\lambda \\
& & & 0 & -J\lambda & \lambda^2 + J^2
\end{pmatrix}
\begin{pmatrix}
\alpha_1 \\ \alpha_2 \\ \alpha_3 \\ \vdots \\ \alpha_{N-3} \\ \alpha_{N-2} \\ \alpha_{N-1}
\end{pmatrix}
=
\begin{pmatrix}
\frac{J}{8} \\ 0 \\ 0 \\ \vdots \\ 0 \\ 0 \\ 0
\end{pmatrix}.
\tag{C.2}
$$

There are many ways of solving such a system of equations, and in this case, we shall simply compute the inverse of the matrix. We suggest this approach because the vector on the right-hand side of the equation only contains a single non-zero value, meaning we have a solution of the form:

$$
\alpha_k = \frac{J}{8} M_{k,1}^{-1},
\tag{C.3}
$$

Where $M$ is our matrix, which is a symmetric Toeplitz tridiagonal matrix that is known to have a analytical inverse [82]. To use this approach, we divide everything by $-J\lambda$ to get the off diagonals to equal 1, giving the new equation:

$$
\begin{pmatrix}
-\frac{\lambda^2+J^2}{J\lambda} & 1 & 0 & & & & & \\
1 & -\frac{\lambda^2+J^2}{J\lambda} & 1 & & & & & \\
0 & 1 & -\frac{\lambda^2+J^2}{J\lambda} & & & & & \\
& & & \ddots & \ddots & \ddots & & \\
& & & & -\frac{\lambda^2+J^2}{J\lambda} & 1 & 0 & \\
& & & & 1 & -\frac{\lambda^2+J^2}{J\lambda} & 1 & \\
& & & & 0 & 1 & -\frac{\lambda^2+J^2}{J\lambda}
\end{pmatrix}
\begin{pmatrix}
\alpha_1 \\ \alpha_2 \\ \alpha_3 \\ \vdots \\ \alpha_{N-3} \\ \alpha_{N-2} \\ \alpha_{N-1}
\end{pmatrix}
=
\begin{pmatrix}
-\frac{1}{8\lambda} \\ 0 \\ 0 \\ \vdots \\ 0 \\ 0 \\ 0
\end{pmatrix},
\quad \text{(C.4)}
$$

and adjusted $\alpha_k$:

$$
\alpha_k = -\frac{1}{8\lambda} M'^{-1}_{k,1}, \quad \text{(C.5)}
$$

where $M'$ is this rescaled matrix. This means we have a diagonal element $D = -\frac{\lambda^2+J^2}{J\lambda}$. Depending on the value of $D$ the form of the substitution required to find the inverse changes:

$$
D = \begin{cases}
2\cosh\omega, & D \geq 2, \\
2\cos\omega, & -2 < D < 2, \\
-2\cosh\omega, & D \leq -2.
\end{cases}
\quad \text{(C.6)}
$$

This gives a result in operators of the new parameter $\omega$. If we limit our parameters to $\lambda > 0$ and $J = \pm 1$ we have the following cases for $D$:

$$
D = \begin{cases}
-\frac{\lambda^2+1}{\lambda}, & J = 1, \\
\frac{\lambda^2+1}{\lambda}, & J = -1.
\end{cases}
\quad \text{(C.7)}
$$

These functions for $\lambda > 0$ have stationary points at $\lambda = 1$, for $J = 1$ this is a maximum value of $-2$, whereas for $J = -1$ this is a minimum value of 2. This then gives us the value of omega for both cases $J = \pm 1$ as:

$$
\omega = \operatorname{arccosh} \frac{\lambda^2+1}{2\lambda}. \quad \text{(C.8)}
$$

However, the inverse of the matrix (and as such the solution to the AGP) is still dependent on the sign of $J$, as expected due to this physically being the difference between ferromagnetic and antiferromagnetic behaviour. We can then write down the first column of the inverse of the matrix as:

$$
M'^{-1}_{k,1} = \begin{cases}
-\frac{\cosh\omega(N+1-k)-\cosh\omega(N-1-k)}{2\sinh\omega\sinh\omega N}, & J = 1, \\[2ex]
(-1)^{k+1}\frac{\cosh\omega(N+1-k)-\cosh\omega(N-1-k)}{2\sinh\omega\sinh\omega N}, & J = -1.
\end{cases}
\quad \text{(C.9)}
$$

This gives us for $\alpha_k$

$$
\alpha_k = \begin{cases}
\frac{\cosh\omega(N+1-k)-\cosh\omega(N-1-k)}{16\lambda\sinh\omega\sinh\omega N}, & J = 1, \\[2ex]
(-1)^{k}\frac{\cosh\omega(N+1-k)-\cosh\omega(N-1-k)}{16\lambda\sinh\omega\sinh\omega N}, & J = -1.
\end{cases}
\quad \text{(C.10)}
$$

This expression is now able to be used but can be simplified further to remove the dependence on $\omega$. The first step is to note that the value of $\omega$ can be expressed as:

$$
\omega = \ln\lambda, \quad \text{(C.11)}
$$

which then gives the same result as in equation (C.8). Using this and the exponential definitions of cosh and sinh we can expand the hyperbolic functions:

$$\cosh \omega (N+1-k) = \frac{\lambda^{2(N+1-k)}+1}{2\lambda^{N+1-k}}, \tag{C.12}$$

$$\cosh \omega (N-1-k) = \frac{\lambda^{2(N-1-k)}+1}{2\lambda^{N-1-k}}, \tag{C.13}$$

$$\sinh \omega = \frac{\lambda^2-1}{2\lambda}, \tag{C.14}$$

$$\sinh \omega N = \frac{\lambda^{2N}-1}{2\lambda^N}. \tag{C.15}$$

Substituting these into the expression for $\alpha_k$ and rearranging gives us the final result, quoted in the main text, for $\alpha_k$:

$$\alpha_k = \begin{cases} \frac{\lambda^{k-1}}{8} \frac{\lambda^{2(N-k)}-1}{\lambda^{2N}-1}, & J=1, \\[2ex] (-1)^k \frac{\lambda^{k-1}}{8} \frac{\lambda^{2(N-k)}-1}{\lambda^{2N}-1}, & J=-1. \end{cases} \tag{C.16}$$

An interesting limit to look at is what happens at the critical point $\lambda \to 1$, which we can compute using L'Hôpital's rule:

$$\lim_{\lambda \to 1} \alpha_k = \begin{cases} \frac{(1)^{k-1}}{8} \frac{2(N-k)(1)^{2(N-k)-1}}{2N(1)^{2N-1}}, & J=1, \\[2ex] (-1)^k \frac{(1)^{k-1}}{8} \frac{2(N-k)(1)^{2(N-k)-1}}{2N(1)^{2N-1}}, & J=-1, \end{cases} \tag{C.17}$$

which can be simplified to:

$$\lim_{\lambda \to 1} \alpha_k = \begin{cases} \frac{1}{8} \frac{N-k}{N}, & J=1, \\[2ex] (-1)^k \frac{1}{8} \frac{N-k}{N}, & J=-1. \end{cases} \tag{C.18}$$

We can then compute the AGP norm by squaring the operator, multiplying by the multiplicity of the type of operator (always $2N$ for ring operators) and sum over $k$, giving (for both $J=1$ and $J=-1$):

$$\begin{aligned} \lim_{\lambda \to 1} ||\mathcal{A}_\lambda||^2 &= 2N \sum_{k=1}^{N-1} \frac{1}{64} \frac{(N-k)^2}{N^2} \\ &= \frac{1}{32N} \left( \sum_{k=1}^{N-1} N^2 - 2\sum_{k=1}^{N-1} Nk + \sum_{k=1}^{N-1} k^2 \right) \\ &= \frac{1}{32N} \left( N^2(N-1) - N^2(N-1) + \frac{N}{6}(N-1)(2N-1) \right) \\ &= \frac{(N-1)(2N-1)}{192}. \end{aligned} \tag{C.19}$$

This shows that exactly at the critical point the norm diverges as $N^2$ compared to at all other values of $\lambda$ where it diverges as $N$ instead.

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
