# Peer review of "A numerical approach for calculating exact non-adiabatic terms in quantum dynamics"

_SciPost Physics, doi:SciPost Phys. 18, 014 (2025)_

## Round 1 · Referee Report · Anonymous (Referee 1) · 2024-3-24

Strengths
Providing a numerical algorithm helps people who want to use counterdiabatic driving.
Weaknesses
Source code is not available.
Report
Requested changes
Optional comments:
1. n->N in Eq.24.
2. The model in Sec.4.1 is the transverse Ising chain with the periodic boundary condition, and the adiabatic gauge potential of it is well studied in the literature. The authors should cite some reference there.
3. The absence of a peak in the adiabatic gauge potential of the Lipkin-Meshkov-Glick model sounds very interesting because we have believed that it shows a significant peak at the critical point. I recommend emphasizing this finding.
4. The present paper is useful for people who want to use counterdiabatic driving. It becomes more useful if the source code (or package code) is available.
Author: Ewen Lawrence on 2024-09-20 [id 4792]
(in reply to Report 1 on 2024-03-24)
Response to Report of Referee 1
We thank the referee for raising helpful and insightful points that we feel the revised version of the manuscript has substantially benefited from. The comments and suggestions made by the referee are addressed below.
Strengths. The manuscript is clearly organised and well written. Providing a numerical algorithm helps people who want to use counterdiabatic driving.
Weaknesses. Simple examples (the ferromagnetic Ising model with a transverse field on some graphs) are only studied.
While we agree that the model we have chosen to use to demonstrate our approach is simple to write down, we disagree that the results we find from this are simple. These models were chosen because of their direct relevance to the current body of work by leading academic groups and companies studying these models using Rydberg atoms, as well as much of the quantum annealing community. There are many problems of interest in optimisation that can be mapped onto this class of Ising models on different graphs, including constrained optimisation problems like QUBO. We have added citations [65-67] to the start of section 4 in the manuscript to try and better show the context in which the example should be considered. We also note that this manuscript is very much focused towards explaining the details of how our method works, rather than exploring the physics of other models. The effectiveness of our technique on these is a good questions for further study.
Source code is not available.
It was always our intention to release source code which implements our method in an open format. At the time of submission this was not quite in a form which would be useful to the community. We have worked on this over the course of the last few months, and have released a repository which includes the source code, please see github.com/ewenlawrence/mAGPy. We now reference this release in the updated version of our manuscript.
Report.
In this manuscript, the authors provide a numerical algorithm for calculating the adiabatic gauge potential. The adiabatic gauge potential is a key idea of nonadiabatic transitions and it can also be used in assisted adiabatic passage, or in other words, short-cuts to adiabaticity by counterdiabatic driving. Their algorithm is based on some previous results, i.e., the variational approach (Ref.25), the algebraic approach (Ref.40), and an idea of nested commutators (Ref.48). They use the above results in an algorithmic way. I believe that the present manuscript is interesting, clearly organised, and well written, and thus it deserves to be published in SciPost Physics.
Requested changes
Optional comments: 1. n → N in Eq.24.
We thank the referee for noticing this typo, which is now labelled as Eq. 35, has now been corrected.
- The model in Sec.4.1 is the transverse Ising chain with the periodic boundary condition, and the adiabatic gauge potential of it is well studied in the literature. The authors should cite some reference there.
The referee is correct to point this out, it is in fact why we start with this model. The Ising model on a chain with periodic boundary conditions allows us to connect directly to prior work with available analytic results. In the previous version of our manuscript we did not give enough details on this previous work and how it connects to our results. We have updated Sec. 4.1 of the new manuscript to give a more thorough discussion of prior work and include additional references, [76,77], of the new manuscript.
- The absence of a peak in the adiabatic gauge potential of the Lipkin-Meshkov-Glick model sounds very interesting because we have believed that it shows a significant peak at the critical point. I recommend emphasising this finding.
The referee is correct that while the manuscript is intended primarily to introduce a new method, the LMG model result is of particular interest. We have expanded the discussion in Sec. 4.3 highlighting the differences between the all-to-all model and the LMG model. We’ve also highlighted this result in the introduction of the manuscript.
- The present paper is useful for people who want to use counterdiabatic driving. It becomes more useful if the source code (or package code) is available.
As discussed above we have now released our code at github.com/ewenlawrence/mAGPy.
Author: Ewen Lawrence on 2024-09-20 [id 4794]
(in reply to Report 3 on 2024-05-04)Response to Report of Referee 3
We thank the referee for raising helpful and insightful points that we feel the revised version of the manuscript has substantially benefited from. The comments and suggestions made by the referee are addressed below.
We thank the referee for this comment, and the suggestions below, which we address as they are raised. We agree that the prior version of our manuscript was not structured in a way that conveyed our key messages that was easy for the reader to follow. We think that the new version has benefited substantially from this restructuring.
The key advantage of our new method is the ability to exploit symmetries and to chose the basis in which the AGP is calculated. We very much build upon the work of the reference pointed to by the referee, as it is there that the insight of the commutator ansatz was introduced. We feel that our new approach and the insights that can be gained from it very much meet the acceptance criteria for SciPost Physics, as we outline in more detail below. We have further emphasised this in section 3 to highlight the advantages of such an approach, and give brief comparisons to other recent methods.
We agree with the referee that Sec.3 is the central part of our manuscript and that the prior version of this section was not as clearly written as it could have been. In the revised version of our manuscript we have substantially reworked and expanded Sec.3 with both this comment, and those of Referee 2 in mind. We have brought the central parts of the discussion that were previously placed in the appendices into this section. We now believe that this new section is far better at conveying the important steps in the method we are introducing. We also provide a more in depth example, showing how the sets of operators are collected and propagated. We include a new simplified flow chart, to give an algorithmic approach to the method in the main text. The section is now fully self contained and does not require the appendix to get understanding of the method. The material which remains in the appendix is solely technical details for a more specialist audience.
We have included more discussion on the key takeaway messages of our manuscript. The new method is mainly advantageous over other methods, e.g., the Krylov approach, as it allows for the basis of the operators to be chosen a priori. As discussed in response to Referee 2 above, this is particularly useful when specific properties of operators want to be studied, e.g., their interaction range, the number of bodies included, etc. In addition to this, it is also important for the implementation of these methods in experiments or on general quantum computing hardware. As it is desirable to avoid a numerically costly decomposition of an operator in one basis into another.
We thank the referee for this comment and we believe we meet the all the criteria. The prior manuscript did not do a good enough job in demonstrating these points, and as such we have updated the the new manuscript accordingly. Below we outline how our manuscript meets each of the required acceptance criteria.
In our work we present a new method that allows for more feasible computation of the AGP in a wide range of models. It gives a novel way of representing the operators which make up the AGP allowing for future work to study the dynamical properties of a wide range of models.
We believe now after a full rework of section 3 alongside smaller changes to wording in other sections to highlight key points, that the work is clear in its meaning.
The reworked section 3 of our manuscript now means that the main text contains all the necessary details to understand our approach, while the technical details in the appendices give a complete picture.
We cite a large body of work, and have continued to add relevant papers in this revised version [57-59,61,65-67,76,77].
At the end of section 3 we provide an overview of the method with a short discussion, alongside in section 4 summarising each of the results for the different graphs studied. This is then finalised in section 5 where we present our conclusions around the promising results shown for the efficiency and speed of our method, alongside pointing out potential next steps with truncation and approximations that can be made.
The introduction and abstract to our paper give a clear overview of our new method and results linking these to the current literature.
As well as the details in the paper we now provide links to an open source codebase which can be used to reproduce all the data in our manuscript.

---

## Round 1 · Referee Report · Anonymous (Referee 2) · 2024-3-31

Strengths
-
The authors provide sufficient details of their approach.
-
The authors apply their method to the interesting system of Ising model on graphs.
Weaknesses
- The authors do not properly review existing methods that are very closely related to the method they introduce.
- The application of their method to more general systems is not discussed.
Report
The main advantage of their method over other methods of a similar nature are unclear. The manuscript will be suitable for the Journal once the authors have considered the changes requested.
Requested changes
I request the authors to address the following question in their manuscript:
Q 1: It is known that the choice of any orthonormal basis (Controlling and exploring quantum systems by algebraic expression of adiabatic gauge potential, PhysRevA.103.012220) will minimize the action. Depending on the problem, some bases are analytically tractable while others are not. The authors employ the Pauli basis for their computation. What are the advantages (in general) in using that basis as compared to say, the Krylov basis?
For context, the Krylov basis approach was developed for the AGP in 1. A Lanczos approach to the Adiabatic Gauge Potential (arXiv:2302.07228) 2. Shortcuts to adiabaticity in krylov space (arXiv:2302.05460) and applied to a large class of problems. It is also known that the Krylov basis in general has lesser number of elements than the Pauli basis. Therefore it seems as if expressing the AGP in the Krylov basis is advantageous over expressing the same in the Pauli basis.
Q 2: The authors say a few words about the presence of a natural truncation point in one of their examples (due to their method). It was shown (arXiv:2302.07228) that the truncation of the Krylov chain connects to the chaoticity of the system.
Can the authors comment on the generic nature of the truncation point in their method vis-à-vis the Krylov method?
Author: Ewen Lawrence on 2024-09-20 [id 4793]
(in reply to Report 2 on 2024-03-31)
Response to Report of Referee 2
We thank the referee for raising helpful and insightful points that we feel the revised version of the manuscript has substantially benefited from. The comments and suggestions made by the referee are addressed below.
Strengths. 1. The authors provide sufficient details of their approach. 2. The authors apply their method to the interesting system of Ising model on graphs.
Weaknesses. 1. The authors do not properly review existing methods that are very closely related to the method they introduce.
The referee is correct to point out that the prior version did not compare our technique to the recently introduced Krylov approach in detail. This was due to our work being largely finished around the time of the release of the two Krylov works arXiv:2302.05460 and arXiv:2302.07228, but final submission was delayed due to personal circumstances.
From our perspective the approach we introduce, the Orthogonal Commutator Expansion (OCE), and the Krylov approach are divergent on their initial aim. We set out with two questions: (1) Can we construct the exact adiabatic gauge potential from the commutator expansion? (2) Can we do this while knowing the operators in the natural basis that will be used in an experiment? Both our method and the Krylov methods will give (1) when possible, although the efficiency between the methods likely varies from model to model, so it is not clear if either has a distinct advantage. The second point is where the two approaches diverge, as the Krylov approach explicitly does not give (2) without a numerically costly decomposition of the obtained Krylov operators into the physical basis of the experiment. The cost of decomposition is likely equal to or less than the cost of the algorithm itself, so this will not fundamentally change the scaling, but does add an extra step which also comes at the expense of not having such an easy grasp on the physical intuition behind the operators obtained.
To perhaps give an example of why the answer to (2) can be important in some scenarios, we can consider the example of implementation of the AGP on real hardware where we need to calculate and implement the CD terms for a spin-1/2 problem beyond low-order expansions. While both the Krylov and the OCE approaches can tackle this, their end result is different. It is much more difficult in an experimental setup based around Pauli gates to implement a particular Krylov operator. It is necessary to decompose the operator into Pauli operators, which is costly to calculate. Whereas the OCE gives the operators in this basis, i.e.~a basis of our choice, naturally.
To summarise, while the Krylov approach will give the AGP in the optimal numerical basis if it can reach convergence, there are scenarios where picking the basis from the start and not have to go through a costly mapping for the operators is advantageous. We therefore don't think that the present manuscript is the correct place for a detailed comparison between the two approaches, and instead include the brief points discussed above into section 3.
- The application of their method to more general systems is not discussed.
We aimed to give an outline of the general approach of our method in Sec.~3. However, with this comment, and comments from Referee 3, it is clear that we did not fully achieve this. We have therefore substantially reworked the text and figures in Sec.~3 and hope that this new version, along with the released source code https://github.com/ewenlawrence/mAGPy, gives the reader a clearer picture of how to apply our method to more general problems.
Report.
The manuscript is appropriately prepared and the methods are explored properly. The main advantage of their method over other methods of a similar nature are unclear. The manuscript will be suitable for the Journal once the authors have considered the changes requested.
Requested changes. I request the authors to address the following question in their manuscript:
Q1: It is known that the choice of any orthonormal basis (Controlling and exploring quantum systems by algebraic expression of adiabatic gauge potential, PhysRevA.103.012220) will minimize the action. Depending on the problem, some bases are analytically tractable while others are not. The authors employ the Pauli basis for their computation. What are the advantages (in general) in using that basis as compared to say, the Krylov basis?
For context, the Krylov basis approach was developed for the AGP in 1. A Lanczos approach to the Adiabatic Gauge Potential (arXiv:2302.07228) 2. Shortcuts to adiabaticity in krylov space (arXiv:2302.05460) and applied to a large class of problems. It is also known that the Krylov basis in general has lesser number of elements than the Pauli basis. Therefore it seems as if expressing the AGP in the Krylov basis is advantageous over expressing the same in the Pauli basis.
As mentioned above we believe that the OCE and Krylov methods are complementary and solve slightly different technical problems. We note that our method is not limited to the Pauli basis, and can be applied to any given trace-orthogonal operator basis. The main advantage of this, is the ability to physically interpret the operators directly.
Note that whilst we represent the operators in a Pauli basis, not all operators are present or are grouped together via symmetries. This means with symmetries applied appropriately, the resulting basis that represents the AGP will be the same size. Additionally by setting a constant basis, we gain the advantage of being able to compare easily between different Hamiltonians defined on the same basis, whereas the Krylov basis is dependent on a particular Hamiltonian, meaning a transformation is required each time to obtain this comparison.
Q 2: The authors say a few words about the presence of a natural truncation point in one of their examples (due to their method). It was shown (arXiv:2302.07228) that the truncation of the Krylov chain connects to the chaoticity of the system. Can the authors comment on the generic nature of the truncation point in their method vis-à-vis the Krylov method?
Our method very naturally lends itself to the language of chaos, by noticing that the Lie algebra generated by the Lie bracket [H, .] with the initial point $\partial_\lambda H$ is exactly the dynamical Lie algebra. There is previous work in the literature that links the size and structure of this Lie algebra to quantum chaos and also the controlability of a system [57-59]. As for a direct comparison to the Krylov method, both should see the same results fundamentally if they truly are finding the minimum degrees of freedom in the system. We have added a short discussion around this in section 3 of our redrafted manuscript.

---

## Round 1 · Referee Report · Anonymous (Referee 3) · 2024-5-4

Strengths
1) The paper demonstrates a clear understanding of the state of the field and will no doubt provide readers with the necessary context to understand the author's perspective
2) The authors introduce a new perspective on known approaches to computing the AGP which should render the problem far more tractable in a number of practical cases
Weaknesses
1) The paper has some structural issues. In particular the exposition of section 3 is both essential to communicating the author's contributions, but is also the weakest part of the paper in terms of presentation. Explaining the structure captured in figure 1 more carefully, perhaps in a fully worked out example, would be a nice change here. As written, I needed to consult other examples and appendices to understand what the authors were saying. A reworking of this section in relation to the rest of the text is probably necessary.
2) The paper is not written in such a way that its novel contributions are obvious. Comparing this paper with, for example, their reference [48], I see that this paper offers clever computational advantages over existing applications of the commutator expansion for the AGP. Is that the extent of the novelty? That the "punchline" contributions of this paper are not entirely clear is certainly a weakness.
If so, in order to meet the criteria of SciPost physics, it is necessary to demonstrate the computational power of these methods more directly
Report
Requested changes
1) A serious rework of section 3, which is essential to the paper's exposition, should be done. As a general rule, I would say that if the reader needs to consult the appendix to understand crucial details, more revisions are necessary. The algebraic structure the authors are discussing at the level of equations should be made clearer in section 3.
2) The authors should state more clearly exactly what the takeaway from this paper should be for the reader. My own sense of how computationally advantageous their formalism is is a bit muddled. This could be cleared up particularly by having a comparison with previously existing numerical methods.
3) This overlaps with point (2), but in restating what the key takeaways are, the authors should keep in mind the journal's acceptance criteria. It is not currently clear that they have been met.
Recommendation
Ask for major revision

---

## Round 2 · Author Response

20 September 2024
Dear Editor, This letter accompanies the resubmission of our manuscript “A numerical approach for calcu- lating exact non-adiabatic terms in quantum dynamics” for publication in SciPost Physics. All of the referees were positive about the novelty and usefulness of our results. Their concerns lie with the lack of detail about the implementation of our method and the lack of source code availability for the reproduction of our results. With the revised version of our manuscript we have addressed these issues as detailed below as well as all of the other concerns raised by the referees.
We hope that with these responses, our manuscript is now suitable for publication. Yours Sincerely,
Ewen D C Lawrence, Sebastian FJ Schmid, Ieva ˇCepait˙e, Peter Kirton and Callum W Duncan

---

## Round 2 · List of Changes

Summary of major changes to the manuscript:
• A complete rewrite of section 3
– Included all necessary equations for using the method into the main text
– Moved the flow chart from the appendix to help clarify the steps in the OCE method
– Included a detailed example of finding the operator basis for a simple 2 site problem.
– Discussion on mathematical structures surrounding the dynamical Lie algebra e.g.
chaos and controlability.
• Updated Section 4.3 on the LMG model to more fully describe the result obtained.
• Supplied links to the source code used to produce all results in form of the mAGPy
package

---

## Editorial Decision

published